# A Multi-Scale and Multi-Level Spectral-Spatial Feature Fusion Network for Hyperspectral Image Classification

**Caihong Mu** [1,*] **, Zhen Guo** [1] **and Yi Liu** [2,*]

[1] Key Laboratory of Intelligent Perception and Image Understanding of Ministry of Education, International Research Center for Intelligent Perception and Computation, Joint International Research Laboratory of Intelligent Perception and Computation, School of Artificial Intelligence, Xidian University, Xi'an 710071, China; zhenguo@stu.xidian.edu.cn

[2] School of Electronic Engineering, Xidian University, Xi'an 710071, China

[*] Correspondence: mucaihongxd@foxmail.com (C.M.); yiliuxd@foxmail.com (Y.L.)

**Abstract:** Extracting spatial and spectral features through deep neural networks has become an effective means of classification of hyperspectral images. However, most networks rarely consider the extraction of multi-scale spatial features and cannot fully integrate spatial and spectral features. In order to solve these problems, this paper proposes a multi-scale and multi-level spectral-spatial feature fusion network (MSSN) for hyperspectral image classification. The network uses the original 3D cube as input data and does not need to use feature engineering. In the MSSN, using different scale neighborhood blocks as the input of the network, the spectral-spatial features of different scales can be effectively extracted. The proposed 3D–2D alternating residual block combines the spectral features extracted by the three-dimensional convolutional neural network (3D-CNN) with the spatial features extracted by the two-dimensional convolutional neural network (2D-CNN). It not only achieves the fusion of spectral features and spatial features but also achieves the fusion of high-level features and low-level features. Experimental results on four hyperspectral datasets show that this method is superior to several state-of-the-art classification methods for hyperspectral images.

**Keywords:** hyperspectral image classification; convolution neural network; multi-scale; 3D–2D alternating residual block

## 1. Introduction

Hyperspectral images (HSIs) have hundreds of continuous spectral bands, and there is a significant correlation between these different bands. At the same time, different types of land-cover in HSIs have different reflections in different spectral bands. Hence, the spectral features of the pixels can be determined according to the spectral reflection values of the pixels at each band. Due to the above characteristics, HSIs are widely used in agricultural planning [1], disaster prevention [2], resource exploration [3], environmental monitoring [4], and other fields. For instance, in the field of resource exploration, abundant geometric spatial information and the spectral information in HSIs can be used to distinguish the characteristics of different substances, which ensures that the objects that cannot be detected in wide band multispectral remote sensing images are able to be detected in HSIs [5]. Thus, improving the classification accuracy of HSIs can improve the accuracy of target detection, make resource detection more accurate, and also make environmental monitoring more comprehensive, which further improves the work efficiency and saves engineering costs. Although HSIs contain a wealth of spatial and spectral information, it is often difficult to obtain enough training samples in practice, leading to "the curse of dimensionality". "The curse of dimensionality" means that the

classification efficiency of HSIs with high dimensions will be reduced under limited training samples. Hence, dimensionality reduction operations are often important in the classification and processing of HSIs. Feature selection and feature extraction are two basic methods used for implementing dimensionality reduction operations. Feature selection [6–8] aims to select a representative spectral band from the original HSI. This method can greatly preserve the physical meaning of the data but may lose a lot of important information. Traditional methods include cluster-based methods [9] and ranking-based methods [10]. Feature extraction aims to extract useful features in HSIs through mathematical transformation, but it will destroy the structural information of the data. Commonly used methods include principal component analysis (PCA) [11,12], independent component analysis (ICA) [13,14], and linear discriminant analysis (LDA) [15]. Whether it is feature selection or feature extraction, it may affect the correlation between the structural information of the HSI and the spectral band to some extent.

To address the above problems, many methods of machine learning have been proposed to improve the classification accuracy on the basis of avoiding the destruction of the original structure information. Traditional HSI classification methods, like support vector machine (SVM) [16,17] and random forest [18], have focused on extracting spectral features of HSIs while ignoring spatial features. As a typical deep learning model, the stacked autoencoder (SAE) [19,20] can extract both spatial and spectral information and then fuse them for HSI classification. Deep belief networks (DBN) [21] and restricted Boltzmann machines [22] have been proposed for combining spatial information and spectral information of HSIs. However, all of the above methods use one-dimensional feature vectors as the input and do not fully utilize the spatial features in HSIs. Considering the above problems, the three-dimensional convolutional neural network (3D-CNN) method [23,24] has selected the neighborhood block as the input of the network model and has simultaneously extracted the spectral and spatial features from the original HSI to obtain better classification accuracy. In order to extract deeper spatial-spectral features, residual learning [25] has been introduced into the convolutional neural network, where the residual network [26–28] helped to train deeper network models and solved the vanishing gradient problem [29,30]. Yang et al. [31] have proposed a two-branch deep convolutional neural network in which one branch is used to extract spectral features and the other branch is used to extract spatial features. A model based on transfer learning has been used to train model parameters and improve the classification performance with limited training samples. However, this method still uses one-dimensional feature vectors as the input when extracting spectral features and does not consider the correlation between different bands. Zhong et al. [32] have proposed a method for classifying HSIs using an end-to-end spectral-spatial residual network (SSRN). The method takes the original 3D cube as input data and does not need to use feature engineering. In the end-to-end spectral-space residual network, the spectral and spatial residual blocks continuously learn recognition features from the rich spectral features and spatial backgrounds in the HSI. The spectral feature obtained by the three-dimensional convolution neural network and the spatial feature obtained by the two-dimensional convolution neural network are fused in a cascade manner. Finally, the fused features are input into the classification layer for HSI classification. However, this method only extracts a single-scale neighborhood block as an input. Single-scale features do not perform well in overall classification accuracy. Song et al. [33] have proposed a deep feature fusion network (DFFN) for HSI classification. The network introduces residual learning to optimize multiple convolution layers as its own mapping. It can simplify the parameters of the network model and is conducive to back propagation. Moreover, the network fuses different levels of output and further improves the classification accuracy. However, this method only combines the output of three levels and the selection of the output layer that needs to be fused was not discussed in detail by the authors. Meng et al. [34] have proposed a new multipath residual network (MPRN) to deal with the problem that deepening network layers result in slowly increasing classification accuracy. The network uses a multipath residual function in parallel instead of stacking multiple residual blocks in the original residual network, making the network wider rather than deeper. A multipath residual network greatly

reduces the redundancy parameters of the network and makes full use of each residual block. However, the network only uses a multipath two-dimensional residual network for feature extraction, which is not sufficient for spectral feature extraction. In addition, most of the existing deep learning models only consider the fusion of spectral and spatial features at the single scale without considering the rich correlation of spectral and spatial features at the multi scale.

Hence, there are still many problems with the existing methods, including insufficient use of scale features, insufficient fusion of spectral and spatial features, and a vanishing gradient problem. To solve these problems and extract sufficient spatial-spectral features from the network, we propose a multi-scale spatial-spectral feature and multi-level feature fusion network (MSSN) for HSI classification. Neighborhood blocks of three different scales are used as the input of the network to extract features of different scales in HSIs. Subsequently, the residual learning block can make full use of the strong correlation of the multi-scale features to extract more discriminant features.

The main contributions of this paper are given below.

(1) Multi-scale feature fusion is carried out by taking three different scales of neighborhood blocks as the input of the network.

(2) The 3D-CNN and the two-dimensional convolution neural network (2D-CNN) are connected in a cascaded way, ensuring that the model can extract more discriminant features by fusing spectral and spatial features. In addition, the proposed 3D–2D alternating residual block fully fuses low-level features and high-level features.

(3) A global average pooling layer is used instead of a full connection layer to reduce model parameters and prevent over-fitting.

The rest of this paper is arranged as follows. Section 2 describes the detailed architecture of our approach. The third section introduces the experiments of MSSN and several state-of-the-art methods on four data sets. Finally, the fourth section provides the conclusion to this paper.

## 2. Methodology

In this section the proposed multi-scale and multi-level spatial-spectral feature fusion network (MSSN) for HSI classification is introduced in detail. All available data are divided into three groups, namely, the training set, testing set, and validation set. Assume a hyperspectral data set $X = \{x_1, x_2, \ldots, x_N\} \in \mathbb{R}^{1 \times 1 \times b}$ where $N$ denotes the number of labeled pixels and $b$ denotes the number of spectral bands. $Y = \{y_1, y_2, \ldots, y_N\} \in \mathbb{R}^{1 \times 1 \times L}$ is the corresponding one-hot label vector, where $L$ denotes the category of objects. The neighborhood blocks of different sizes are cut from the pixels in $X$ and a new data set $Z = \{z_1, z_2, \ldots, z_N\} \in \mathbb{R}^{H \times H \times L}$ is established, where $H \times H$ represents the size of the neighborhood blocks. To fully extract the spectral and multi-scale features of HSIs, three 3D cubes of different scale neighborhood blocks are taken as the input of the MSSN network, which can be expressed as a training set, $Z^1 = \{Z_1^1, Z_2^1, Z_3^1\}$, validation set, $Z^2 = \{Z_1^2, Z_2^2, Z_3^2\}$, and testing set, $Z^3 = \{Z_1^3, Z_2^3, Z_3^3\}$, respectively. Their corresponding one-hot label vector sets are $Y^1 = \{Y_1^1, Y_2^1, Y_3^1\}$, $Y^2 = \{Y_1^2, Y_2^2, Y_3^2\}$, and $Y^3 = \{Y_1^3, Y_2^3, Y_3^3\}$, respectively. The MSSN network trains itself hundreds of epochs through training set $Z^1$ and its corresponding one-hot label vector set $Y^1$. In each epoch, the MSSN network updates network parameters using $Z^1$ and $Y^1$. Then, the temporary model generated by the network is monitored by the validation set $Z^2$ and its corresponding label vector set $Y^2$. In other words, the validation set is used to test the temporary model. If the classification accuracy (named the validation rate here) is higher than before, the parameters of the new temporary model will be recorded. The best model with the highest validation rate will be obtained after all the training epochs are finished. Finally, the testing set $Z^3$ is used to test and evaluate the best model. Figure 1 shows the flow chart of the procedure by which MSSN deals with HSI classification.

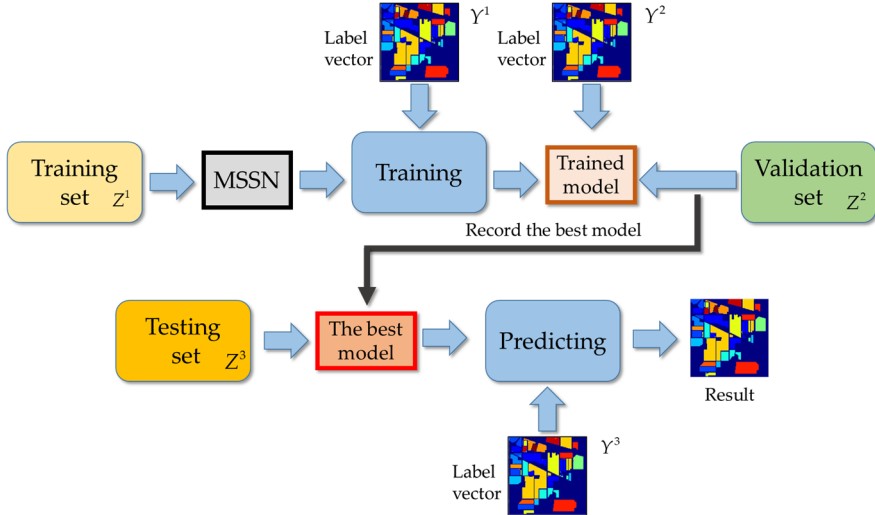

**Figure 1.** The flow chart of the multi-scale and multi-level spectral-spatial feature fusion network (MSSN) dealing with hyperspectral image (HSI) classification.

## 2.1. Refining Spectral and Spatial Features via Continuous 3D–2D Alternating Residual Blocks

The CNN model has been widely used in HSI classification and has achieved good results. However, with the increase in network layers, the classification accuracy does not increase but decreases. There are two reasons for this phenomenon. Firstly, using a small number of samples to train complex network models results in an over-fitting phenomenon which leads to the reduction of classification accuracy. Secondly, when using the back-propagation method to calculate derivatives, with the increase in the depth of the network, the amplitude value of the gradient of back propagation (from the output layer to the initial layers of the network) is sharply reduced. As a result, the derivative of the whole loss function with respect to the weights of the first several layers is very small. In this way, when the gradient descent method is used, the weights of the initial layers change very slowly, meaning they cannot learn features from the samples effectively. However, by adding an identity shortcut connection between layers to construct residual blocks, the problem of precision reduction can be reduced [35]. For this reason, we designed a 3D–2D alternating residual block to extract spectral and spatial features from the original three-dimensional HSI cube continuously. The residual block connection method not only solves the vanishing gradient problem but also fully fuses the spectral features extracted by 3D-CNN with the spatial features extracted by 2D-CNN. As shown in Figure 2, the 3D convolution layer and the 2D convolution layer are alternately connected in the form of a residual. Such a structure can be conducive to the back propagation of the gradient in the network, thus improving the classification accuracy while solving the vanishing gradient problem.

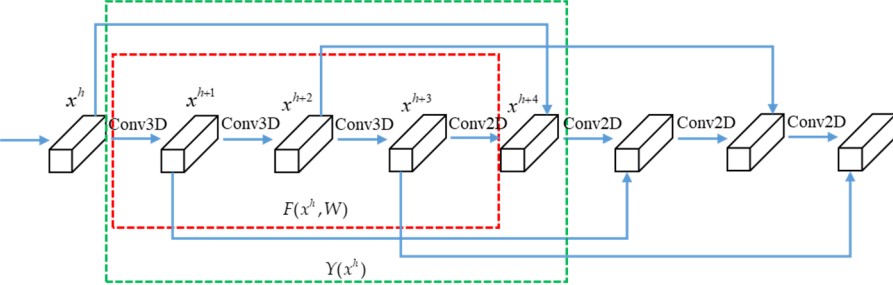

**Figure 2.** 3D–2D alternating residual block. Legend: Conv3D, 3D convolution; Conv2D, 2D convolution.

In the 3D–2D alternating residual block shown in Figure 2 we take the first alternating residual block as an example to introduce the proposed 3D–2D alternating residual block. In Figure 2, $x^h$, $x^{h+1}$,

$x^{h+2}$, and $x^{h+3}$ represent the four characteristic tensors obtained by 3D-CNN, and $x^{h+4}$ represents the first characteristic tensor obtained by 2D-CNN. By a padding strategy, the space size of feature tensors $x^{h+1}$, $x^{h+2}$, and $x^{h+3}$ are kept the same and are unchanged. Then, the size of the first feature tensor $x^h$ obtained by the 3D-CNN is converted to be the same as the size of the first feature tensor $x^{h+4}$ obtained by the 2D-CNN, which is conducive to self-mapping. The output $Y(x^h)$ of the 3D–2D alternating residual block can be expressed as

$$Y(x^h) = F(x^h, W) + x^h, \tag{1}$$

where $W$ is the convolution kernel and $F(x^h, W)$ is a residual function constructed by three 3D convolution layers and one 2D convolution layer, which can be expressed as

$$F(x^h, W) = R\big(R\big(R\big(R\big(x^h * W^1 + b^1\big) * W^2 + b^2\big) * W^3 + b^3\big) * W^4 + b^4\big), \tag{2}$$

$W^1$, $W^2$, and $W^3$ and $b^1$, $b^2$, and $b^3$ are the weights and biases of the first three three-dimensional convolution layers in the 3D–2D alternating residual block. $W^4$ and $b^4$ are the weights and biases of the first two-dimensional convolution layer. $R(\cdot)$ is the activation function of the rectified linear unit (ReLU).

In addition, we use a batch normalization (BN) operation on each convolution layer to standardize the learning process of each convolution operation. This strategy can make the training process of the network model more effective. The BN operations are defined as

$$BN(x^n) = \frac{x^n - E(x^n)}{\sqrt{Var(x^n) + \varepsilon}} \cdot \gamma + \beta, \tag{3}$$

$x^n$ denotes the input of layer $n$ before the BN operation. $BN(x^n)$ denotes the output of layer $n$ after the BN operation. $E(\cdot)$ and $Var(\cdot)$ denote the expected function and variance function of the input characteristic tensor, respectively. $\gamma$ and $\beta$ are learnable parameter vectors and $\varepsilon$ is a parameter for numerical stability.

### 2.2. Fusion of Multi-Scale Features and Optimization by Cross-Entropy Loss

In the proposed MSSN we use $7 \times 7$, $11 \times 11$, and $15 \times 15$ neighborhood blocks of different scales as the input, and extract spatial-spectral features using 3D–2D alternating residual blocks. The three scales of the feature tensors are still $7 \times 7$, $11 \times 11$, and $15 \times 15$. To facilitate the fusion of features we use the transition layer to make the features of the three scales become the same size, i.e., $7 \times 7$. The transition layer consists of two $3 \times 3$ convolution layers and pooling layers and the same padding operation is used to ensure the uniformity of size. Fusion operations can be represented as

$$M = C_1(F_1) \oplus C_1(F_2) \oplus C_2(F_3), \tag{4}$$

where $M$ denotes the fusion tensor of the multi-scale features; $C_1$, $C_2$, and $C_3$ denote three different convolution and pooling operations of the transition layer to ensure that the size of the three features are consistent; $\oplus$ denotes the concatenate operation, which fuses the multi-scale spatial-spectral features after processing; and $F_1$, $F_2$, and $F_3$ represent the three scale features obtained by the proposed 3D–2D alternating residual block. The obtained multi-scale and multi-level spatial-spectral fusion features are vectorized at the global average pooling layer and then transferred to the softmax layer for final classification.

After constructing the deep learning model, the model trains hundreds of epochs through training set $Z^1$ and its corresponding one-hot label vector set $Y^1$. In the training process, the parameters of

MSSN are updated by back propagation and cross-entropy is set as the objective function, which is defined as

$$Loss(\hat{y}, y) = \sum_{i=1}^{L} y_i \left( \log \sum_{j=1}^{L} e^{\hat{y}_j} - \hat{y}_i \right), \tag{5}$$

where $\hat{y} = \{\hat{y}_1, \hat{y}_2, \ldots, \hat{y}_L\}$ is the predictive vector and $y = \{y_1, y_2, \ldots, y_L\}$ is the ground-truth label vector. The validation set $Z^2$ is used to monitor the temporary model generated by the network to judge the classification performance of the model and record the best model. Finally, the testing set $Z^3$ is used to test the recorded best model and to evaluate the generalization ability of the MSSN model.

### 2.3. MSSN for HSI Classification

Considering the inadequate use of scale features by existing technologies, we propose to use multi-scale inputs to train the MSSN model and obtain multi-scale features by fusion. HSIs are rich in spectral and spatial features so we propose a deep learning framework which can extract spatial features from HSIs successively and alternately, as shown in Figure 3. Compared with a simple CNN, the MSSN fully fuses the spectral features extracted by 3D-CNN with the spatial features extracted by 2D-CNN by introducing a continuous 3D–2D alternating residual block which improves the classification accuracy. We take the Indian Pines hyperspectral data set as an example to describe the proposed MSSN.

In our proposed MSSN model, $7 \times 7 \times 200$, $11 \times 11 \times 200$, and $15 \times 15 \times 200$ hyperspectral neighborhood blocks are used as inputs to extract multiple scale features from HSIs. Taking the $7 \times 7 \times 200$ neighborhood block as an example, in the first convolutional layer, we use 24 3D convolution kernels of size $1 \times 1 \times 20$ to convolve the input neighborhood block under the $(1, 1, 20)$ step and get 24 feature cubes of size $7 \times 7 \times 10$. Because HSIs are rich in spectral features, we use 24 3D convolution kernels of size $1 \times 1 \times 20$, which allows the convolution kernel to focus more on spectral features and quickly reduce dimensions. The resulting feature cube then passes through the 3D–2D alternating residual block, which alternately and in an orderly fashion connects the spectral features extracted by the 3D-CNN and the spatial features extracted by the 2D-CNN in the form of a residual. In the 3D-CNN part of the 3D–2D alternating residual block we use 24 3D convolution kernels of size $1 \times 1 \times 3$ to ensure the extraction of rich spectral features. For the last convolutional layer of this part, we use 24 3D convolution kernels of size $1 \times 1 \times 10$ to guarantee the input size of the 2D-CNN portion of the 3D–2D alternating residual block. In the 2D-CNN portion of the 3D–2D alternating residual block, we firstly use 240 2D convolution kernels of size $3 \times 3 \times 24$ to connect 3D-CNN and 2D-CNN. Then, we use 240 2D convolution kernels of size $3 \times 3 \times 240$ to extract spatial features in the hyperspectral, which facilitate residual operations. Finally, we use 24 2D convolution kernels of size $3 \times 3 \times 240$ to reduce the dimensions of the feature map. The 3D–2D alternating residual block combines high-level features and low-level features while merging spatial features and spectral features. For example, our proposed 3D–2D alternating residual block combines the low-level spectral features extracted by the first 3D convolutional layer with the high-level spatial features extracted by the first 2D convolutional layer in the form of residuals. The fusion of the spatial-spectral features and the multi-level features is realized at the same time.

Because the three kinds of neighborhood blocks still have different scales after the same 3D–2D alternative residual operation, and in order to facilitate fusion, we transfer the feature tensor obtained from the 3D–2D alternating residual block to a transition layer so that the output features of the three scales are all $7 \times 7 \times 24$. The transition layer consists of two convolution layers, each of which contains 24 2D convolution kernels of size $3 \times 3 \times 24$ and two $(3, 3)$ max pooling layers. In all the 3D–2D alternating residual blocks and transition layers we use a padding operation to ensure the output size.

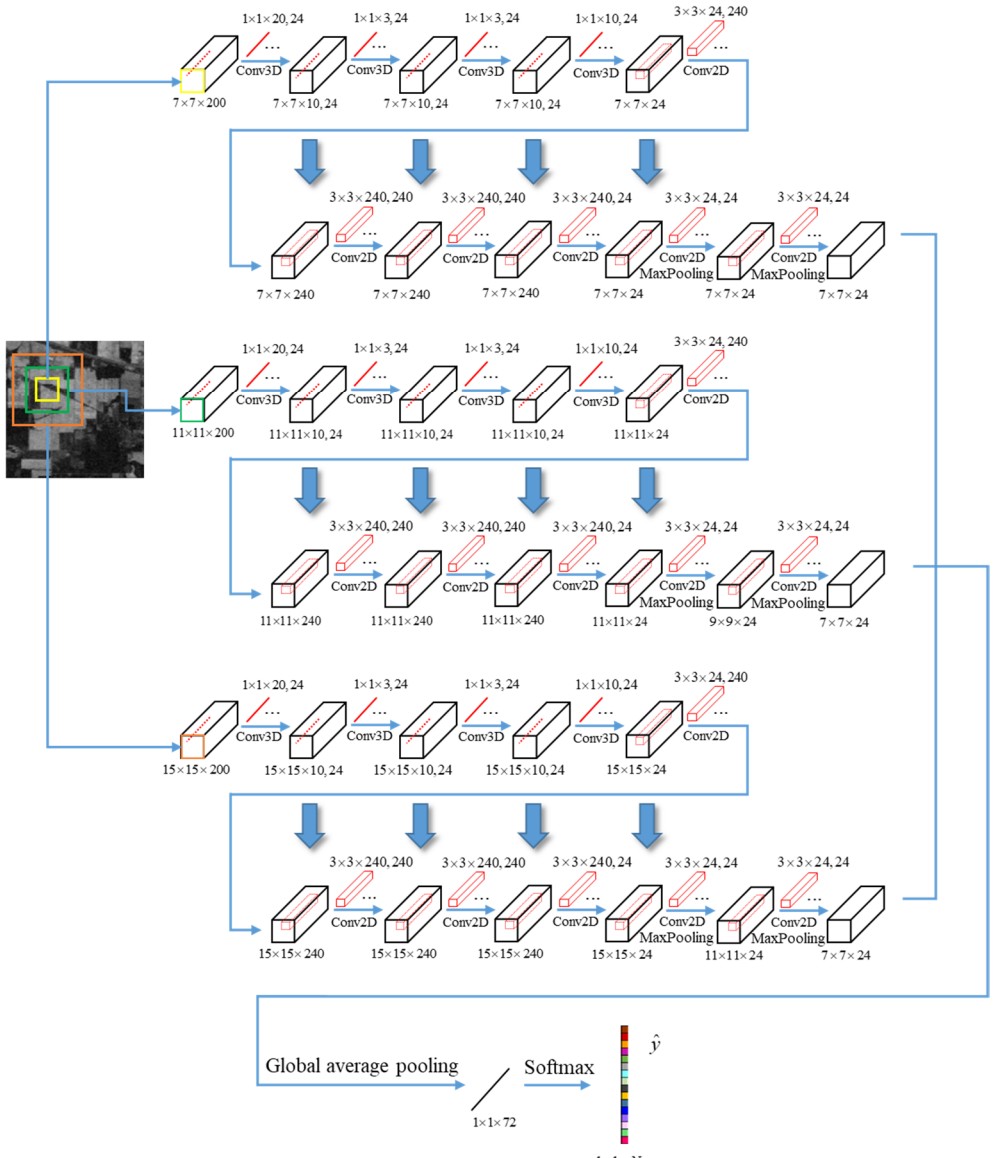

**Figure 3.** MSSN with $7 \times 7 \times 200$, $11 \times 11 \times 200$, and $15 \times 15 \times 200$ hyperspectral neighborhood blocks as input; each scale branch of the network contains a 3D–2D alternating residual block and a transition layer. The global average pooling layer and softmax layer transform $7 \times 7 \times 72$ fusion features into a $1 \times 1 \times N$ output feature vector $\hat{y}$.

Through the above two parts the features of the three scales are fused. We use the global average pooling layer instead of the full connection layer to transform multi-scale and multi-level spatial-spectral fusion features of size $7 \times 7 \times 72$ into feature vectors of size $1 \times 1 \times 72$, which greatly reduces the model parameters and prevents the occurrence of an over-fitting phenomenon. Then, the softmax layer generates the prediction vector set $\hat{Y} = \{\hat{y}_1, \hat{y}_2, \ldots, \hat{y}_N\} \in \mathbb{R}^{1 \times 1 \times L}$ according to the land-cover category of the HSI. In MSSN, the BN operation is used after each convolution layer to enhance the classification performance of the model.

## 3. Experimental Results

### 3.1. Experimental Data Sets

Four real hyperspectral datasets, including Indian Pines, Pavia University, Salinas scene, and Kennedy Space Center (KSC) datasets, were used to demonstrate the effectiveness of our proposed method. These datasets are publicly accessible online [36,37].

(1) Indian Pines data set: the Indian Pines data set is the result of the acquisition of a remote sensing experimental area in the northwest of the Indian state by Airborne Visible Infrared Imaging Spectrometer (AVIRIS). The dataset is a remote sensing image 145 pixels in width and height and with a spatial resolution of 20 meters per pixel. It has 220 wavelengths ranging from 0.4 μm to 2.5 μm. After 20 noise bands are removed, the remaining 200 bands are used in the experiment. The Indian Pines data set contains 16 types of land-cover and 10,249 samples. In Figure 4, (a) is a false-color image of the Indian Pines data set and (b) is a ground-truth map of the Indian Pines data set.

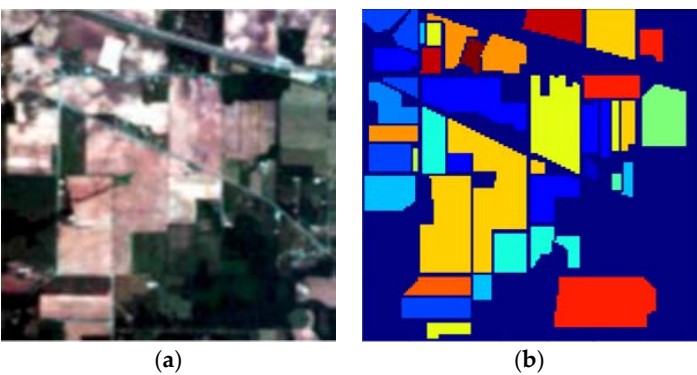

(**a**)                    (**b**)

**Figure 4.** Indian Pines data set. (**a**) False-color image of the Indian Pines data. (**b**) Ground-truth map of the Indian Pines.

(2) Pavia University data set: the Pavia University data set is the data collected by Reflective Optics System Imaging Spectrometer (ROSIS) from Pavia University in northeastern Italy. The dataset is 340 pixels wide and 610 pixels high, with a spatial resolution of 1.3 m per pixel. It has 115 wavelengths ranging from 0.43 μm to 0.86 μm. After 12 noise bands were removed, the remaining 103 bands were used in the experiment. The data set of Pavia University contains nine types of land-cover and 42,776 samples. In Figure 5, (a) is a False-color image of the Pavia University data set and (b) is a ground-truth map of the Pavia University data set.

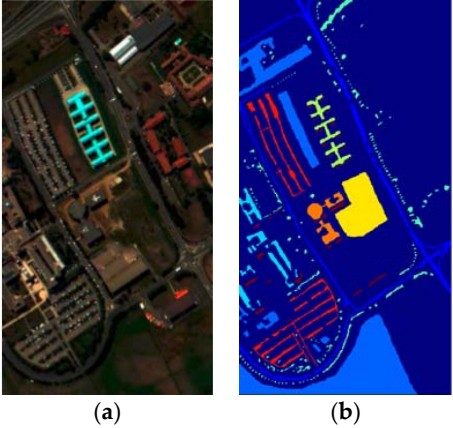

(**a**)                    (**b**)

**Figure 5.** Pavia University data set. (**a**) False-color image of Pavia University. (**b**) Ground-truth map of Pavia University.

(3) Salinas scene data set: the Salinas scene data set is collected by the AVIRIS sensor over Salinas Valley, California. The data set is a remote sensing image 217 pixels wide and 512 pixels high which has a spatial resolution of 3.7 meters per pixel and contains 224 bands. After 20 water absorption bands were removed, the remaining 204 bands were used in the experiment. The Salinas scene data set contains a total of 16 types of land-cover and 54,129 samples. In Figure 6, (a) is a false-color image of the Salinas scene data set and (b) is a ground-truth map of the Salinas scene data set.

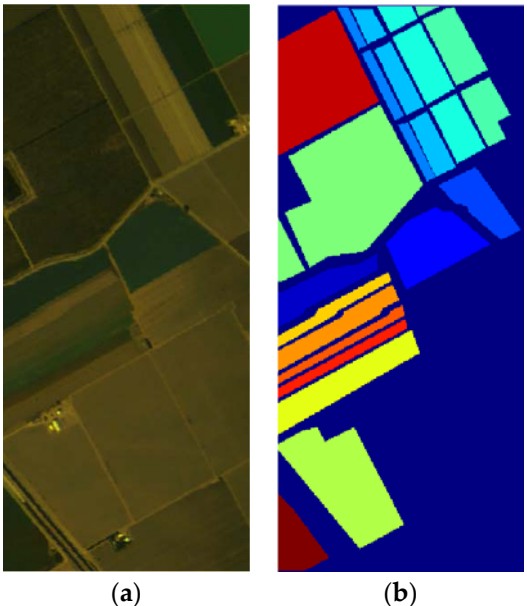

(**a**)  (**b**)

**Figure 6.** Salinas scene data set. (**a**) False-color image of the Salinas scene data. (**b**) Ground-truth map of the Salinas scene.

(4) KSC data set: the KSC data set is collected by AVIRIS over the KSC, Florida. The data set is a remote sensing image 614 pixels wide and 512 pixels high, with a spatial resolution of 18 meters per pixel. After removing water absorption bands and bands with a low signal-noise ratio, the remaining 176 band data were used in the experiment. The KSC dataset contains a total of 13 types of land-cover and 5211 samples. In Figure 7, (a) is a false-color image of the KSC data set and (b) is a ground-truth map of the KSC data set.

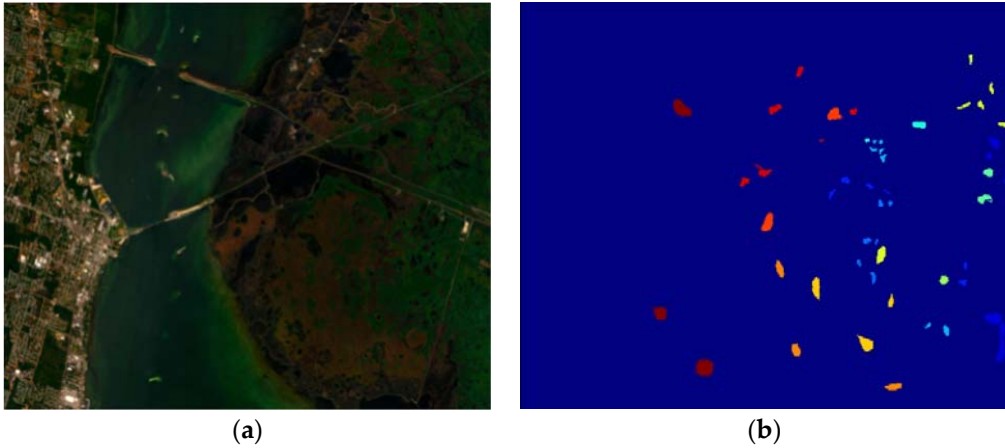

(**a**)  (**b**)

**Figure 7.** Kennedy Space Center (KSC) data set. (**a**) False-color image of the KSC data. (**b**) Ground-truth map of the KSC.

Table 1 introduces a summary of each class of land-cover in four data sets and the number of samples each contains.

**Table 1.** Information regarding samples of each class in four data sets.

| | | Indian Pines | | | Salinas Scene | |
|---|---|---|---|---|---|---|
| No. | Color | Class | Samples | Color | Class | Samples |
| 1 | | Alfalfa | 46 | | Brocoli_green_weeds_1 | 2009 |
| 2 | | Corn-notill | 1428 | | Brocoli_green_weeds_2 | 3726 |
| 3 | | Corn-mintill | 830 | | Fallow | 1976 |
| 4 | | Corn | 237 | | Fallow_rough_plow | 1394 |
| 5 | | Grass-pasture | 483 | | Fallow_smooth | 2678 |
| 6 | | Grass-trees | 730 | | Stubble | 3959 |
| 7 | | Grass-pasture-mowed | 28 | | Celery | 3579 |
| 8 | | Hay-windrowed | 478 | | Grapes_untrained | 11,271 |
| 9 | | Oats | 20 | | Soil_vinyard_develop | 6203 |
| 10 | | Soybean-notill | 972 | | Corn_senesced_green_weeds | 3278 |
| 11 | | Soybean-mintill | 2455 | | Lettuce_romaine_4wk | 1068 |
| 12 | | Soybean-clean | 593 | | Lettuce_romaine_5wk | 1927 |
| 13 | | Wheat | 205 | | Lettuce_romaine_6wk | 916 |
| 14 | | Woods | 1265 | | Lettuce_romaine_7wk | 1070 |
| 15 | | Buildings-grass-trees-drives | 386 | | Vinyard_untrained | 7268 |
| 16 | | Stone-steel-towers | 93 | | Vinyard_vertical_trellis | 1807 |
| | | Pavia University | | | KSC | |
| No. | Color | Class | Samples | Color | Class | Samples |
| 1 | | Asphalt | 6631 | | Scrub | 761 |
| 2 | | Meadows | 18,649 | | Willow Swamp | 243 |
| 3 | | Gravel | 2099 | | CP Hammock | 256 |
| 4 | | Trees | 3064 | | CP Oak | 252 |
| 5 | | Painted metal sheets | 1345 | | Slash Pine | 161 |
| 6 | | Bare Soil | 5029 | | Oak Broadleaf | 229 |
| 7 | | Bitumen | 1330 | | Hardwood Swamp | 105 |
| 8 | | Self-blocking Bricks | 3682 | | Graminoid Marsh | 431 |
| 9 | | Shadows | 947 | | Spartina Marsh | 520 |
| 10 | | | | | Cattail Marsh | 404 |
| 11 | | | | | Salt marsh | 419 |
| 12 | | | | | Mud Flats | 503 |
| 13 | | | | | Water | 927 |

### 3.2. Experimental Setup

For each data set the samples were divided into a training set, validation set, and testing set. The training set was used to update network parameters. The validation set was used to monitor the temporary model generated by the network and retain the model with the highest validation rate. The testing set was used to evaluate the classification performance of the preserved model. Among these, for the Indian Pines and KSC datasets, we randomly selected 10%, 10%, and 80% samples of each type to form the training set, validation set, and testing set, respectively. For the Pavia University and Salinas scene datasets, we randomly selected 5%, 5%, and 90% samples from each class to form the training set, validation set, and testing set, respectively.

In order to evaluate the classification performance of the proposed method we used the overall accuracy (OA), average accuracy (AA), and Kappa coefficient as the evaluation index [38]. We used the average of five experimental results as the final result.

In our experiment, after appropriate experimental adjustment, the train epoch was set to 200 times, the batch size was set to 16, the learning rate was set to 0.0001, and the momentum of the BN operation was set to 0.8. All experiments were carried out on an NVIDIA 1080ti graphics card using Python language.

### 3.3. Influence of Parameters

#### 3.3.1. The Effectiveness of Multi-Scale Inputs

In order to verify the validity of the multi-scale idea in this paper, we compared the experimental results of single-scale and multi-scale inputs on the four data sets proposed. As shown in Table 2, the experiment was carried out under the input scale conditions of $7 \times 7$, $11 \times 11$, $15 \times 15$, and multi-scale fusion proposed in this paper. The network structure of each single-scale input ($7 \times 7$ or $11 \times 11$ or $15 \times 15$) was the same as that of a single branch in MSSN. The data set selection was as described above, and the effects of different scales on the OA, AA, and Kappa coefficient were able to be observed.

**Table 2.** The results of different scales on different data sets. Data are given as mean ± standard deviation. Legend: OA, overall accuracy; AA, average accuracy.

| Indian Pines (10%) | $7 \times 7$ | $11 \times 11$ | $15 \times 15$ | $7 \times 7$, $11 \times 11$, $15 \times 15$ |
|---|---|---|---|---|
| OA (%) | 96.89 ± 0.42 | 98.16 ± 0.14 | 98.46 ± 0.13 | 99.12 ± 0.15 |
| AA (%) | 97.70 ± 0.25 | 96.84 ± 0.91 | 97.65 ± 0.65 | 99.23 ± 0.07 |
| Kappa × 100 | 96.44 ± 0.35 | 97.90 ± 0.54 | 98.24 ± 0.18 | 98.99 ± 0.18 |
| **Pavia University (5%)** | $7 \times 7$ | $11 \times 11$ | $15 \times 15$ | $7 \times 7$, $11 \times 11$, $15 \times 15$ |
| OA (%) | 98.91 ± 0.17 | 99.61 ± 0.12 | 99.76 ± 0.11 | 99.94 ± 0.02 |
| AA (%) | 99.11 ± 0.24 | 99.53 ± 0.19 | 99.64 ± 0.14 | 99.93 ± 0.05 |
| Kappa × 100 | 99.36 ± 0.14 | 99.49 ± 0.16 | 99.69 ± 0.15 | 99.92 ± 0.04 |
| **Salinas Scene (5%)** | $7 \times 7$ | $11 \times 11$ | $15 \times 15$ | $7 \times 7$, $11 \times 11$, $15 \times 15$ |
| OA (%) | 99.39 ± 0.21 | 99.46 ± 0.12 | 99.78 ± 0.08 | 99.84 ± 0.11 |
| AA (%) | 99.52 ± 0.18 | 99.50 ± 0.15 | 99.64 ± 0.09 | 99.88 ± 0.05 |
| Kappa × 100 | 99.33 ± 0.25 | 99.49 ± 0.13 | 99.76 ± 0.19 | 99.82 ± 0.12 |
| **KSC (10%)** | $7 \times 7$ | $11 \times 11$ | $15 \times 15$ | $7 \times 7$, $11 \times 11$, $15 \times 15$ |
| OA (%) | 98.21 ± 0.13 | 99.22 ± 0.17 | 99.46 ± 0.09 | 99.69 ± 0.12 |
| AA (%) | 98.65 ± 0.24 | 99.09 ± 0.25 | 99.04 ± 0.21 | 99.54 ± 0.23 |
| Kappa × 100 | 98.12 ± 0.32 | 99.17 ± 0.25 | 99.39 ± 0.18 | 99.65 ± 0.15 |

From the results we can see that, first of all, when selecting small-scale blocks such as the $7 \times 7$ neighborhood blocks as the input, in most cases the AA of the experimental results is greater than the OA. When a large-scale block such as the $15 \times 15$ neighborhood block is selected as the input, the OA of the experimental results is greater than the AA. For example, the experimental results regarding the Indian Pines and KSC data sets reflect this well. This indicates that small-scale blocks have a better classification effect on small and discrete sample categories and that large-scale blocks have a better classification effect on all the data, but also that neighborhood blocks which are too large may introduce noise when extracting features of categories that contain fewer samples. Secondly, from Table 2, it can be seen that the results of multi-scale inputs are superior to those of the single-scale inputs in three evaluation indexes, showing that multi-scale inputs can generate multi-scale features. Compared with the single-scale features generated by the single scale inputs, these features have a richer correlation between spatial structure information and texture information. The idea of scale feature fusion is used to combine the advantages of the small scale and the large scale, which not

only improves the overall classification performance of the network but also focuses on the feature extraction of small samples. The experimental results of four different data sets verify the validity of the multi-scale feature fusion idea.

### 3.3.2. The Selection of the Number of 3D–2D Residual Connections in the MSSN

In the 3D–2D alternating residual block proposed above, the 3D-CNN is used to extract spectral features of HSIs, the 2D-CNN is used to extract spatial features of HSIs, and, finally, the spatial-spectral features and multi-level features are fused by a 3D–2D alternating residual block. In this section the number of 3D–2D alternate residual connections is determined by experiments. From Table 3, N represents the number of 3D–2D alternate residual connections. Different classification accuracies can be obtained by changing the number of 3D–2D alternating residual connections and the proper number can be finally determined by classification accuracy. The results are as shown in Table 3.

**Table 3.** The overall accuracy (%) obtained by the MSSN with different numbers of 3D–2D residual connections. Data are given as mean ± standard deviation.

| N | Indian Pines | Pavia University | Salinas Scene | KSC |
|---|---|---|---|---|
| 1 | 97.52 ± 0.56 | 98.22 ± 0.23 | 97.46 ± 0.45 | 96.69 ± 1.24 |
| 2 | 98.58 ± 0.24 | 99.15 ± 0.05 | 98.64 ± 0.25 | 98.67 ± 0.19 |
| 4 | **99.12 ± 0.15** | **99.94 ± 0.02** | 99.84 ± 0.11 | **99.69 ± 0.12** |
| 6 | 99.05 ± 0.12 | 99.77 ± 0.09 | **99.86 ± 0.08** | 99.42 ± 0.17 |

It can be seen from Table 3 that when the number of 3D–2D alternating residual connections is 4, the overall classification accuracy on the four data sets is the best. When the number is 1 or 2, the network cannot fully extract and fuse the spatial and spectral features of the HSIs. When we continue to increase the number of 3D–2D alternant residual connections on the basis of 4, it greatly increases the complexity of the model and causes an over-fitting phenomenon in the training, which reduces the classification accuracy to a certain extent. In the Salinas scene data set, six 3D–2D alternating residual connections were found to be slightly higher than four 3D–2D alternating residual connections. The main reason for this is that Salinas scene data set contains a large number of samples, and each kind of sample is relatively concentrated, which is conducive to learning and suppresses the occurrence of the over-fitting phenomenon. According to the experimental results, we chose four 3D–2D alternating residual connections to construct our proposed 3D–2D alternating residual block.

### 3.3.3. The Selection of the Learning Rate in the MSSN

As an important super parameter in deep learning, the learning rate determines whether and when the objective function can converge to the local minimum. The appropriate learning rate can make the objective function converge to the local minimum value in the appropriate time. In this subsection, we chose learning rates of 0.01, 0.001, 0.0001, 0.00003, and 0.00001 to train the MSSN model by using four hyperspectral datasets under the condition that other environments remain unchanged. Figure 8 shows a line chart of the overall accuracy of the four hyperspectral datasets when selecting different learning rates.

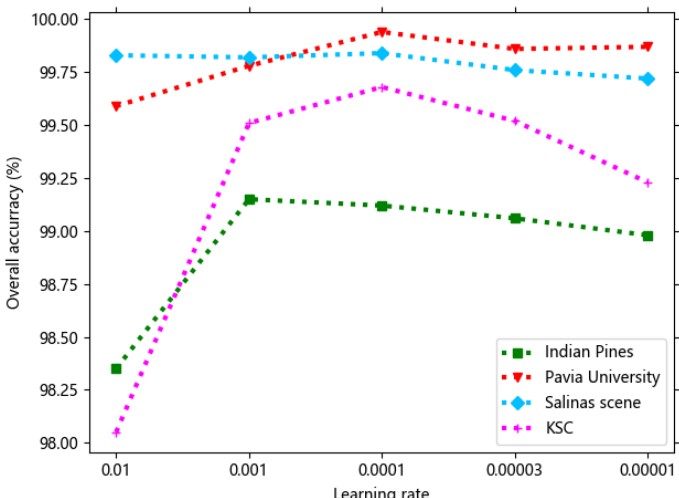

**Figure 8.** Overall accuracy of the four hyperspectral datasets when selecting different learning rates in the MSSN.

It can be observed from Figure 8 that, in most cases, the polyline corresponding to learning rates ranging 0.01 to 0.0001 shows an increasing trend. When the learning rate is smaller than 0.0001, the polyline tends to be flat or even decrease. This is because when the learning rate is too large, the objective function may skip the local optimal solution in the process of convergence. When the learning rate is too small, the convergence will be slow, which leads to the failure of the objective function to converge to the local optimal solution. It can also cause the model to overfit, which leads to the reduction of accuracy. Hence, we chose a learning rate of 0.0001 as the super parameter used in our experiments.

### 3.4. Classification Results of Hyperspectral Datasets

We compared the proposed method with SVM [15] and several state-of-the-art methods: 3D-CNN [22], ResNet [24], SSRN [31], DFFN [32], and MPRN [33].

SVM is a traditional and classical machine learning method which can be used for classification. 3D-CNN extracts spectral and spatial features from HSIs simultaneously using three-dimensional convolution kernels. ResNet uses the self-mapping idea to extract rich features from HSIs, which is beneficial to back propagation. SSRN combines the spectral features obtained by three-dimensional convolution and the spatial features obtained by two-dimensional convolution in a cascade manner, which ensures that the model can continuously extract spectral features and spatial features. DFFN combines the features extracted by ResNet in different levels for classification. MPRN proposes the use of a wider residual network instead of a deeper one for feature extraction. In order to make a fair comparison, we adjusted the model parameters of these comparison methods to their best state and trained them in their same experimental environment.

(1) Classification of the Indian Pines data set: Table 4 gives the classification results of various methods obtained from the Indian Pines data set in terms of three evaluation indicators, namely, OA, AA, and Kappa. Figure 9 shows the ground-truth map of the Indian Pines dataset and the classification map of the seven algorithms on the Indian Pines dataset. Figure 10 shows a line chart of the overall accuracy of the seven algorithms when selecting different percentages of training samples.

**Table 4.** Classification results obtained by different methods on the Indian Pines data set. Data are given as mean ± standard deviation. Legend: SVM, support vector machine; 3D-CNN, three-dimensional convolutional neural network; SSRN, spectral-spatial residual network; DFFN, deep feature fusion network; MPRN, multipath residual network.

| Class | SVM | 3D-CNN | ResNet | SSRN | DFFN | MPRN | MSSN |
|---|---|---|---|---|---|---|---|
| 1 | 83.33 | 59.52 | 94.44 | 97.37 | **100** | **100** | 97.14 |
| 2 | 72.78 | 91.60 | 96.03 | 99.35 | 98.88 | 99.39 | **99.41** |
| 3 | 65.19 | 87.01 | 98.22 | 97.62 | 99.41 | 97.93 | **99.45** |
| 4 | 63.08 | 85.98 | 95.36 | 79.98 | **100** | 98.97 | 97.93 |
| 5 | 90.57 | 88.51 | 95.04 | 98.98 | 98.43 | 95.30 | **100** |
| 6 | 95.59 | 98.93 | 99.65 | 98.66 | **99.99** | 99.47 | 99.14 |
| 7 | 69.23 | 84.62 | 100 | 90.48 | 100 | 100 | **100** |
| 8 | 93.51 | 100 | 100 | 100 | 100 | 100 | **100** |
| 9 | 72.22 | 94.44 | 66.67 | **94.44** | 80 | 93.33 | 88.24 |
| 10 | 71.09 | 84 | 98.45 | 97.64 | 99.51 | **99.74** | 97.51 |
| 11 | 86.11 | 91.04 | 95.64 | 98.66 | 98.18 | 97.82 | **99.2** |
| 12 | 72.28 | 76.03 | 98.59 | 98.32 | 98.15 | 98.36 | **98.91** |
| 13 | 96.76 | 99.46 | 96.43 | 100 | 100 | 100 | **100** |
| 14 | 97.89 | 97.54 | 99.71 | 98.33 | 100 | 100 | **100** |
| 15 | 47.71 | 77.87 | 85.64 | **100** | 99.98 | 97.71 | 99.67 |
| 16 | 80.95 | 97.62 | 95.83 | **98.65** | 95.83 | 97.22 | 91.02 |
| OA (%) | 81.18 ± 1.54 | 90.38 ± 0.89 | 96.92 ± 0.59 | 98.22 ± 0.25 | 99.06 ± 0.16 | 98.71 ± 0.21 | **99.12 ± 0.15** |
| AA (%) | 78.64 ± 1.49 | 88.39 ± 1.46 | 96.73 ± 0.85 | 98.08 ± 0.59 | 98.24 ± 0.50 | 98.34 ± 0.60 | **99.23 ± 0.07** |
| Kappa × 100 | 78.36 ± 1.02 | 89.03 ± 0.96 | 96.49 ± 0.57 | 97.97 ± 0.85 | 98.93 ± 0.23 | 98.34 ± 0.45 | **99.02 ± 0.18** |

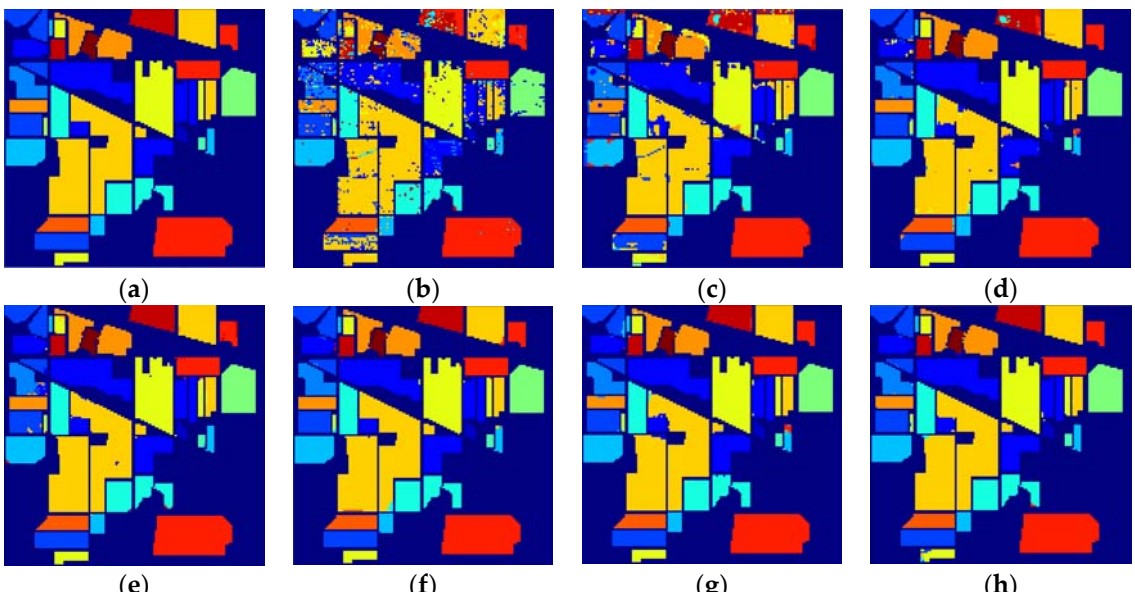

**Figure 9.** Classification maps for the Indian Pines data set. (**a**) Ground truth. (**b**) SVM. (**c**) 3D-CNN. (**d**) ResNet. (**e**) SSRN. (**f**) DFFN. (**g**) MPRN. (**h**) MSSN.

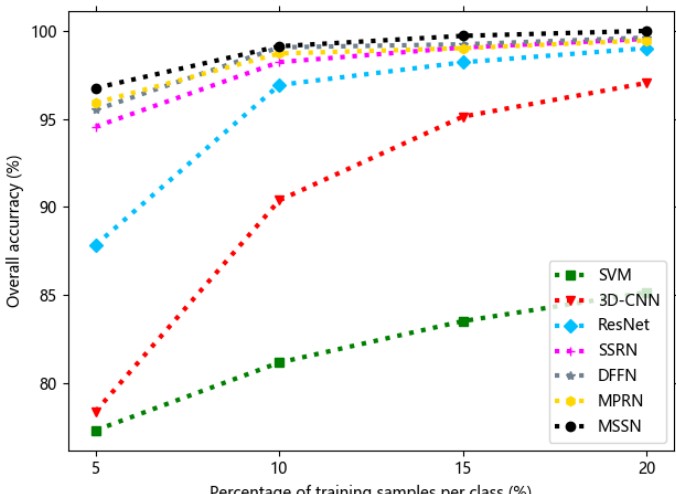

**Figure 10.** Overall classification accuracies obtained by different methods when considering different percentages of training samples on the Indian Pines data set.

From Table 4, it can be seen that the OA result of MSSN is higher than that of the classical methods SVM, 3D-CNN, and ResNet by 17.94%, 8.74%, and 2.2%, respectively. MSSN also outperforms the most advanced methods SSRN, DFFN, and MPRN by 0.9%, 0.06%, and 0.41% in terms of OA, respectively. It can be observed from Figure 9 that the classification results of SVM and 3D-CNN have an obvious "phenomenon of salt and pepper". The classification results of both DFFN and our MSSN are most similar to the ground truth.

(2) Classification of the Pavia University data set: Table 5 lists the classification results of various methods on the three evaluation indicators OA, AA and Kappa. Figure 11 shows the ground-truth map of the Pavia University data set and the classification map of the seven algorithms on the Pavia University data set. Figure 12 shows a line chart of the overall accuracy of the seven algorithms when selecting different percentages of training samples.

**Table 5.** Classification results obtained by different methods on the Pavia University data set. Data are given as mean ± standard deviation.

| Class | SVM | 3D-CNN | ResNet | SSRN | DFFN | MPRN | MSSN |
|---|---|---|---|---|---|---|---|
| 1 | 92.87 | 95.84 | 98.81 | 99.94 | 99.78 | 99.89 | **99.98** |
| 2 | 98.09 | 98.79 | 99.95 | 100 | 100 | 99.98 | **100** |
| 3 | 74.03 | 87.42 | 92.93 | 94.21 | 98.79 | **99.68** | 99.1 |
| 4 | 94.68 | 96.39 | 99.64 | 98.26 | 96.55 | 98.76 | **99.96** |
| 5 | 99.37 | 99.14 | 99.83 | 99.51 | 99.01 | 99.42 | **100** |
| 6 | 85.72 | 91.27 | 96.20 | 100 | 100 | 100 | **100** |
| 7 | 83.07 | 93.35 | 96.57 | 99.58 | 99.49 | **100** | 99.92 |
| 8 | 90.74 | 95.77 | 95.98 | 99.43 | 98.46 | 99.45 | **99.91** |
| 9 | 99.78 | 97 | 99.88 | 100 | 90.49 | 98.59 | **100** |
| OA (%) | 93.38 ± 0.68 | 96.26 ± 0.18 | 98.51 ± 0.28 | 99.51 ± 0.09 | 99.27 ± 0.16 | 98.71 ± 0.25 | **99.94 ± 0.02** |
| AA (%) | 90.93 ± 0.54 | 94.99 ± 0.80 | 98.19 ± 0.75 | 99.49 ± 0.17 | 98.63 ± 0.52 | 98.34 ± 0.33 | **99.93 ± 0.05** |
| Kappa × 100 | 91.18 ± 0.82 | 95.03 ± 0.50 | 98.03 ± 0.31 | 97.97 ± 0.12 | 99.03 ± 0.19 | 98.34 ± 0.32 | **99.92 ± 0.04** |

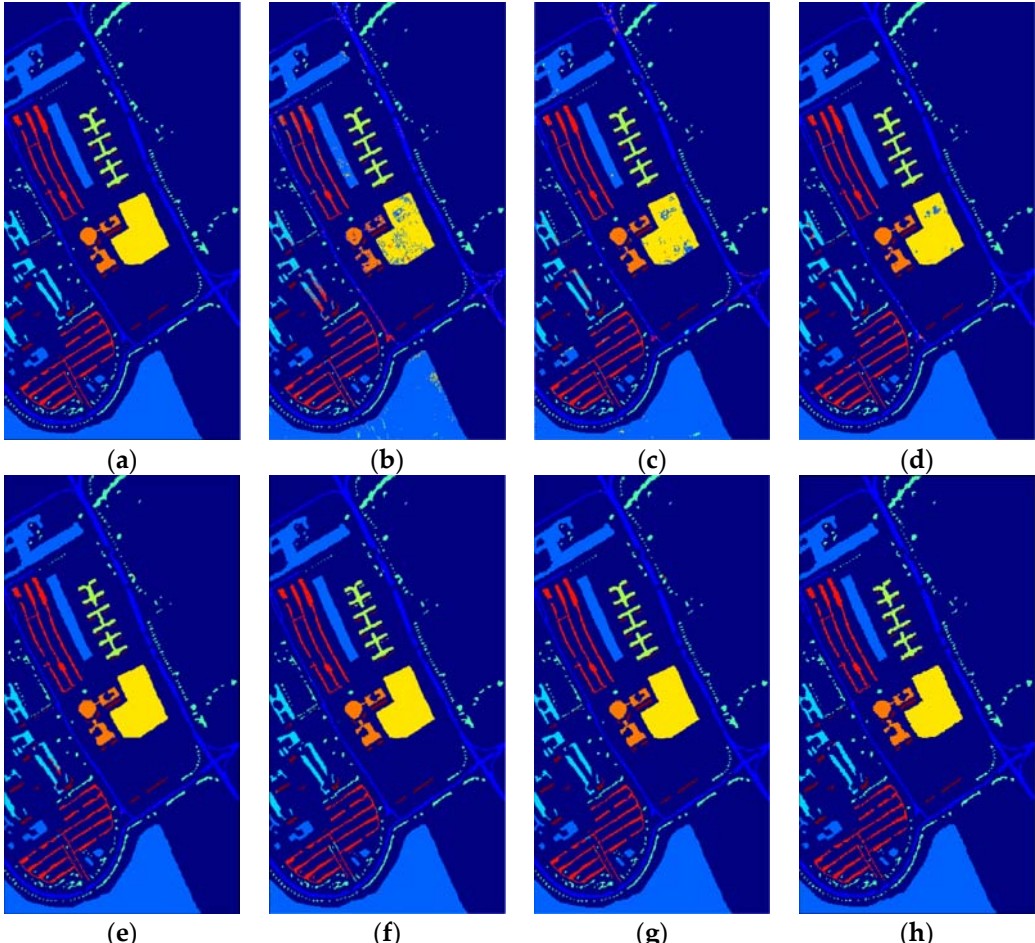

**Figure 11.** Classification maps for the Pavia University data set. (**a**) Ground truth. (**b**) SVM. (**c**) 3D-CNN. (**d**) ResNet. (**e**) SSRN. (**f**) DFFN. (**g**) MPRN. (**h**) MSSN.

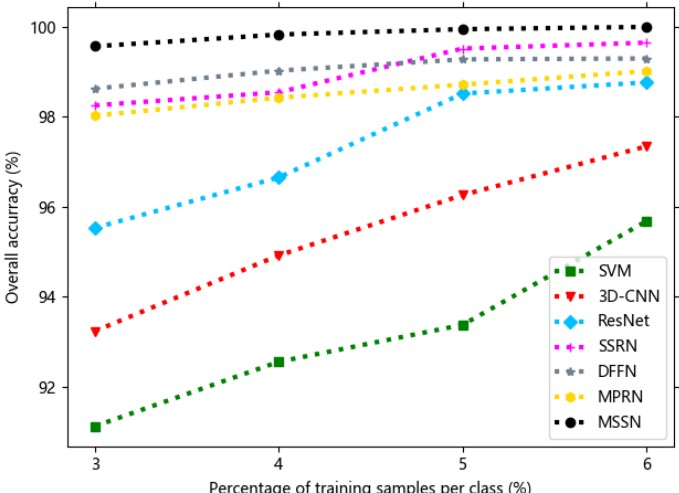

**Figure 12.** Overall classification accuracies obtained by different methods when considering different percentages of training samples on the Pavia University data set.

It can be seen from Table 5 that the OA result of MSSN is higher than that of the classical methods SVM, 3D-CNN, and ResNet by 6.56%, 3.68%, and 1.43%, respectively. MSSN also outperforms the most advanced methods SSRN, DFFN, and MPRN by 0.43%, 0.67%, and 1.23% in terms of OA, respectively. It can also be found that MSSN obtains the best classification result for almost all classes compared

with other methods. From Figure 11, it can be observed that the classification results of SVM and 3D-CNN also have an obvious "phenomenon of salt and pepper". The classification results of SSRN, DFFN, MPRN, and the proposed MSSN are very similar to the ground truth.

(3) Classification of the Salinas scene data set: Table 6 lists the classification results of various methods on the three evaluation indicators OA, AA, and Kappa. Figure 13 shows the ground-truth map of the Salinas scene data set and the classification map of the seven algorithms on the Salinas scene data set. Figure 14 shows a line chart of the overall accuracy of the seven algorithms when selecting different percentages of training samples.

**Table 6.** Classification results obtained by different methods on the Salinas scene data set. Data are given as mean ± standard deviation.

| Class | SVM | 3D-CNN | ResNet | SSRN | DFFN | MPRN | MSSN |
|---|---|---|---|---|---|---|---|
| 1 | 96.8 | 95.29 | 99.83 | 100 | 100 | 100 | **100** |
| 2 | 97.01 | 99.80 | 100 | 100 | 100 | 100 | **100** |
| 3 | 99.31 | 99.57 | 100 | 100 | 100 | 100 | **100** |
| 4 | 98.04 | 97.58 | 99.87 | 99.84 | 99.52 | 99.36 | **99.91** |
| 5 | 96.58 | 99.84 | 99.75 | 99.29 | **99.96** | 99.75 | 98.3 |
| 6 | 95.93 | 99.04 | 100 | 100 | 100 | 100 | **100** |
| 7 | 97.03 | 97.94 | 99.93 | **100** | **100** | **100** | 99.97 |
| 8 | 82.96 | 93.68 | 91.86 | 98.39 | 98.73 | 99.49 | **99.92** |
| 9 | 98.79 | 98.61 | 99.85 | 100 | 100 | 100 | **100** |
| 10 | 87.12 | 94.09 | 97.15 | **100** | 99.39 | **100** | 99.73 |
| 11 | 91.23 | 94.98 | 99.07 | 99.16 | 97.51 | 99.48 | **99.79** |
| 12 | 98.96 | 99.62 | 100 | 100 | 99.54 | 100 | **100** |
| 13 | 93.69 | 96.9 | 100 | 100 | 100 | 99.88 | **100** |
| 14 | 85.55 | 94.99 | 99.47 | 98.76 | 99.37 | **100** | 99.69 |
| 15 | 69.04 | 87.31 | 91.38 | 95.72 | 98.33 | 98.37 | **99.76** |
| 16 | 89.34 | 95.34 | 99.94 | 99.69 | 99.87 | 100 | **100** |
| OA (%) | 89.35 ± 0.43 | 95.56 ± 0.79 | 96.91 ± 0.25 | 99.01 ± 0.23 | 99.38 ± 0.08 | 99.63 ± 0.16 | **99.84 ± 0.11** |
| AA (%) | 92.34 ± 0.36 | 96.54 ± 0.64 | 98.31 ± 0.35 | 99.57 ± 0.15 | 99.51 ± 0.19 | 99.79 ± 0.09 | **99.88 ± 0.05** |
| Kappa × 100 | 88.09 ± 0.71 | 95.06 ± 0.51 | 96.55 ± 0.48 | 98.89 ± 0.31 | 99.31 ± 0.11 | 99.59 ± 0.06 | **99.82 ± 0.12** |

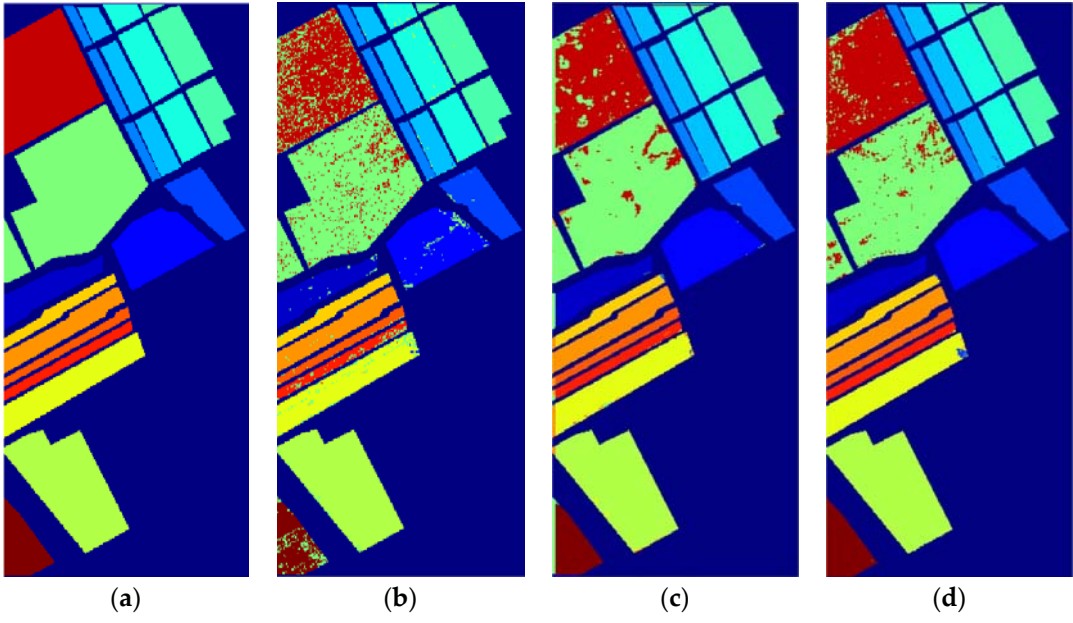

(**a**)          (**b**)          (**c**)          (**d**)

**Figure 13.** *Cont.*

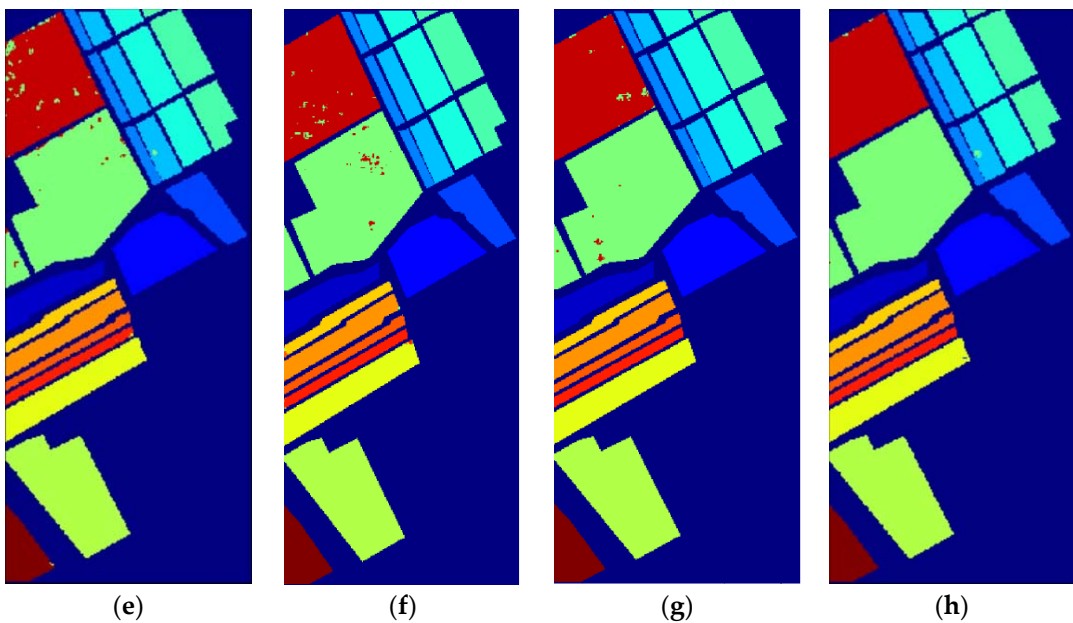

**Figure 13.** Classification maps for the Salinas scene data set. (**a**) Ground truth. (**b**) SVM. (**c**) 3D-CNN. (**d**) ResNet. (**e**) SSRN. (**f**) DFFN. (**g**) MPRN. (**h**) MSSN.

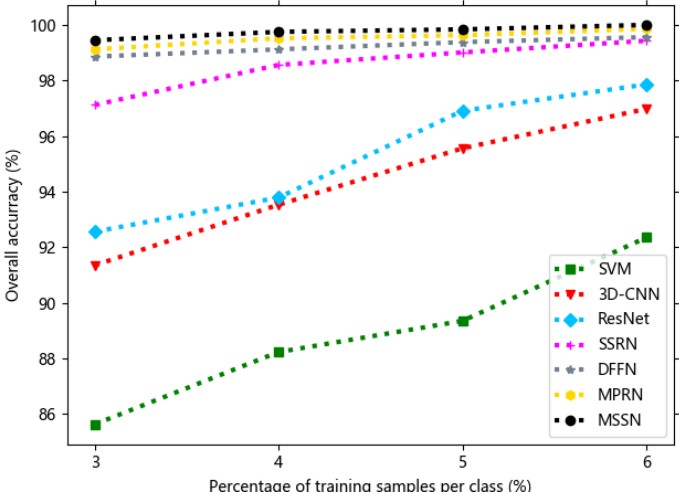

**Figure 14.** Overall classification accuracies obtained by different methods when considering different percentages of training samples on the Salinas scene data set.

From Table 6, it can be seen that the OA result of MSSN is higher than that of the classical methods SVM, 3D-CNN, and ResNet by 10.49%, 4.28%, and 2.93%, respectively. MSSN also outperforms the most advanced methods SSRN, DFFN, and MPRN by 0.83%, 0.46%, and 0.21% in terms of OA, respectively. It can be observed that class 8 (Grapes_untrained) and class 15 (Vinyard_untrained) are difficult to classify, while MSSN achieves the best effect. It can be found from Figure 13 that the classification results generated by the proposed MSSN are most similar to the ground truth with the best regional consistence.

(4) Classification of the KSC data set: Table 7 lists the classification results of various methods on the three evaluation indicators OA, AA, and Kappa. Figure 15 shows the ground-truth map of the KSC data set and the classification map of the seven algorithms on the KSC data set. Figure 16 shows a line chart of the overall accuracy of the seven algorithms when selecting different percentages of training samples.

**Table 7.** Classification results obtained by different methods on the KSC data set. Data are given as mean ± standard deviation.

| Class | SVM | 3D-CNN | ResNet | SSRN | DFFN | MPRN | MSSN |
|---|---|---|---|---|---|---|---|
| 1 | 95.18 | 97.81 | 97.26 | 99.84 | 98.19 | 100 | **100** |
| 2 | 86.31 | 87.21 | 94.57 | 100 | 100 | 100 | **100** |
| 3 | 83.55 | 95.24 | 97.38 | 94.76 | **100** | **100** | 99.01 |
| 4 | 73.57 | 63.87 | 75.74 | 95.05 | 96.91 | 93.81 | **100** |
| 5 | 59.31 | 80.69 | 82.81 | 87.5 | **100** | 99.24 | 91.41 |
| 6 | 69.57 | 87.92 | 85.64 | 97.34 | 100 | 100 | **100** |
| 7 | 92.63 | 92..63 | 91.76 | 100 | 100 | 72.62 | **100** |
| 8 | 93.81 | 97.68 | 97.37 | 100 | 99.71 | 100 | **100** |
| 9 | 98.08 | 96.79 | 100 | 100 | 100 | 100 | **100** |
| 10 | 91.21 | 94.23 | 100 | 100 | 100 | 100 | **100** |
| 11 | 95.77 | 98.41 | 100 | 100 | 100 | 100 | **100** |
| 12 | 91.83 | 95.36 | 99.01 | 98.76 | 100 | 99.49 | **100** |
| 13 | 99.28 | 96.29 | 100 | 100 | 100 | 100 | **100** |
| OA (%) | 91.18 ± 0.59 | 93.63 ± 0.47 | 96.39 ± 0.37 | 98.87 ± 0.22 | 99.56 ± 0.09 | 99.08 ± 0.19 | **99.68 ± 0.12** |
| AA (%) | 86.93 ± 0.86 | 91.09 ± 0.70 | 93.01 ± 0.20 | 98.14 ± 0.57 | 99.23 ± 0.25 | 98.46 ± 0.21 | **99.54 ± 0.23** |
| Kappa × 100 | 90.17 ± 1.15 | 92.92 ± 0.62 | 95.98 ± 0.29 | 98.74 ± 0.26 | 99.52 ± 0.12 | 98.98 ± 0.37 | **99.65 ± 0.15** |

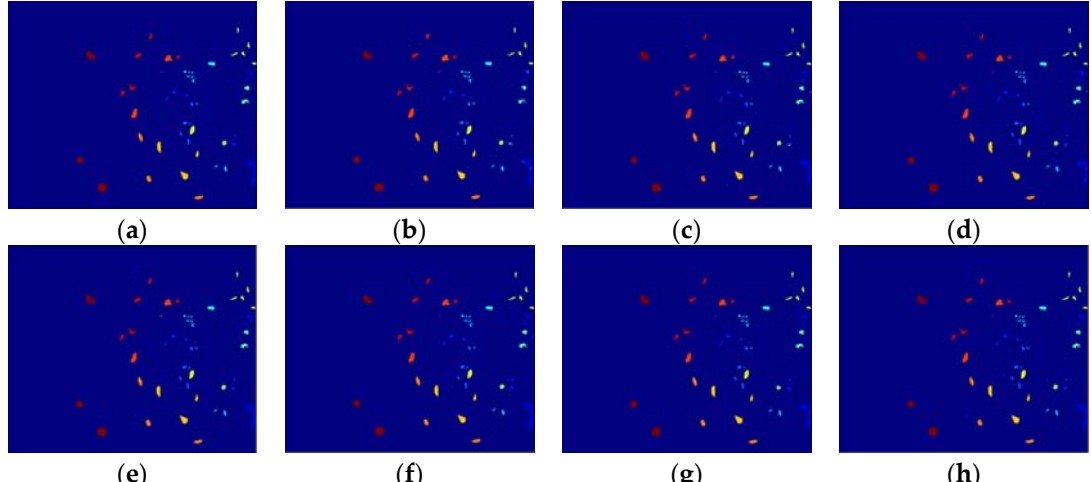

**Figure 15.** Classification maps for the KSC data set. (**a**) Ground truth. (**b**) SVM. (**c**) 3D-CNN. (**d**) ResNet. (**e**) SSRN. (**f**) DFFN. (**g**) MPRN. (**h**) MSSN.

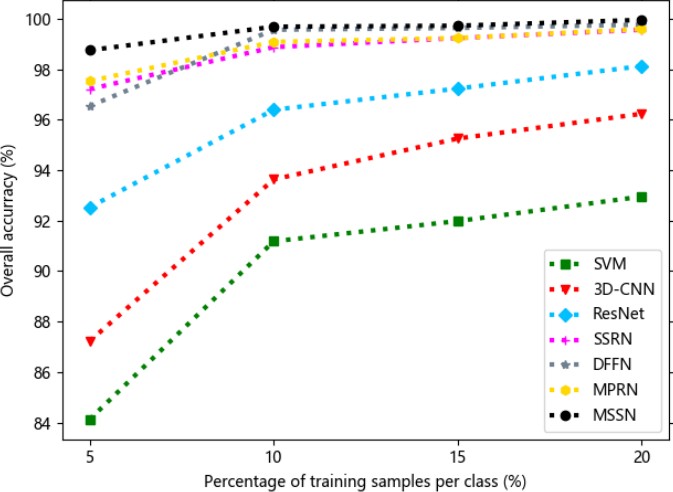

**Figure 16.** Overall classification accuracies obtained by different methods when considering different percentages of training samples on the KSC data set.

From Table 7, it can be seen that the OA result of MSSN is higher than that of the classical methods SVM, 3D-CNN, and ResNet by 8.5%, 6.05%, and 3.29%, respectively. MSSN also outperforms the most advanced methods SSRN, DFFN, and MPRN by 0.81%, 0.12%, and 0.6% in terms of OA, respectively. We can observe that the proposed MSSN obtains the best classification result for almost every class compared with other methods.

Overall, MSSN achieved the best classification performance in most categories and the proposed MSSN achieved the best classification results in terms of OA, AA, and Kappa. There are three main reasons for this performance improvement: (1) 3D-CNN and 2D-CNN are connected in a cascade way, which ensures that the model can continuously extract spectral and spatial features; (2) the multi-scale idea effectively retains the correlation and complementarity between different scales, and integrates the advantages of each scale; (3) in the special 3D–2D alternating residual block, low-level features and high-level features are fused, which makes the model easier to train. Compared with SSRN, which simply combines spatial and spectral information, MSSN introduces the idea of extracting multi-scale features, which makes full use of the advantages of each scale and integrates the rich correlation and complementarity between each scale. Compared with DFFN, MSSN does not use any feature engineering to preprocess the original data, which greatly retains the original spatial structure information. To some extent, feature engineering will make the processed data lose the spatial information of the original image and further affect the classification effect.

In order to test the generalization ability and robustness of the proposed MSSN for different training samples, we randomly selected 5%, 10%, 15%, and 20% of the labeled samples as the training data of the Indian Pines and KSC data sets and selected 3%, 4%, 5%, and 6% of the labeled samples as the training data of the Pavia University and Salinas scene data sets. It can be seen from Figures 10, 12, 14 and 16 that when the training data is limited, MSSN can still maintain a high classification accuracy compared with other single traditional classical methods such as SVM and 3D-CNN, etc. Compared with other more advanced complex networks such as DFFN and MPRN, the classification results of MSSN are better than other state-of-the-art methods using different amounts of training data.

## 4. Conclusions

In this paper a new MSSN deep neural network model was proposed to extract more abundant features from HSIs and classification. Compared with other existing network models, MSSN is composed of special 3D–2D alternating residual blocks. A three-dimensional convolution layer and a two-dimensional convolution layer are alternately connected in the form of a residual, which can effectively extract and fuse the spatial and spectral information in HSIs. In addition, compared with the single-scale network model, MSSN proposes the idea of multi-scale feature fusion, which makes full use of the advantages of each scale and greatly retains the rich correlation and complementarity between different scales. At the same time, the proposed MSSN model can achieve high classification accuracy without any feature engineering, which greatly retains the spatial-spectral information of the original image and makes our method more generalized. Finally, the proposed 3D–2D alternating residual block can extract more high-level features for classification, and is conducive to back-propagation, meaning the network model can still maintain a higher classification accuracy in deeper networks. The experimental results show that the classification results of this method are better than other state-of-the-art methods for the four datasets. Using fewer training samples, this method can still maintain good classification performance, which further verifies that the proposed MSSN has good robustness and generalization ability.

Compared with the classical methods, such as SVM, 3D-CNN, and ResNet, the proposed MSSN shows overwhelming superiority. Compared with some recent methods, such as DFFN, MPRN, and SSRN, our MSSN also shows significant advantages. MSSN solves the problem of insufficient spectral-spatial feature fusion in DFFN. In addition, compared with MPRN and SSRN, the idea of the multi scale is proposed in MSSN, fully integrating the scale features of HSIs and making MSSN solve

classification more effectively. Finally, the proposed method solves the problem of gradient vanishing by using a 3D–2D alternating residual block, which often happens in the deep neural network.

Although multi-scale input brings about a more accurate classification ability, the three-branch network model also increases the complexity of calculation. In future work we will study a more lightweight deep learning network model for feature extraction and fusion which can still obtain a higher classification accuracy.

**Author Contributions:** Conceptualization, Z.G.; formal analysis, Z.G.; funding acquisition, C.M.; investigation, Z.G.; methodology, Z.G.; project administration, C.M.; software, Z.G.; supervision, C.M.; validation, C.M.; visualization, Z.G.; writing—original draft, Z.G.; writing—review & editing, C.M. and Y.L. All authors have read and agreed to the published version of the manuscript.

**Funding:** This research was funded by the National Natural Science Foundation of China under grants 61672405, 61876141, U1701267, 61772399, 61773304, and 61802295.

**Acknowledgments:** The authors would like to thank the editor and anonymous reviewers who handled our paper.

**Conflicts of Interest:** The authors declare no conflict of interest.

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
