# Peer review of "A Multi-Scale and Multi-Level Spectral-Spatial Feature Fusion Network for Hyperspectral Image Classification"

_remotesensing, doi:10.3390/rs12010125_

Round 1

Reviewer 1 Report

This is a very interesting work, the authors proposed a new method for hyperspectral image classification. The article is well written and organised, and the proposed method has also achieved a good performance. I recommend a minor revision before accepting for publication. Please find my comments below,

The authors did review past studies and pointed out the problems. However, they failed to summarize them. In order to make readers feel the importance of your study, a summary of current research problems is needed in this section. The authors did highlight their contributions to this study. But there are so many listed in the article, I suggest the authors summarize them and reorganize them. In order to help readers to understand your methodology, I suggest adding a flow chart of the proposed new method in this section. Experimental results. There are too many tables used in introducing the dataset, you can try to combine them or simply them. The authors did report research results clearly, but more explanation is needed to help the reader understand the research results. Conclusions should be a separate section. A discussion section is needed to address the contributions of your study. Moreover, you also need to compare your work with past studies to highlight your contribution. It is one of the most important sections if the article. In addition to summarize your research results, the limitation of your study and potential future research direction should also be included in the article.

Some minor comments as following

Line 103 table 1. Everyone knows DEM, it’s not necessary to have a footnote here to give the full name of the DEM. Otherwise, you should also give the full name for other factors, such as NDVI. The format of the text part, figure and table need to be double-checked to satisfy the journal’s requirements.

Reviewer 2 Report

Technically, it is good paper. However if we use hyperspectral image for agriculture planning, disaster prevention, resource exploration and environmental monitoring, how detailed information we need? Please provide some information.

What are the purpose to improve the classification accuracy?

Research method can be improved for easy to understand.

In lines 326 – 328, please explain why you set up these numbers.

Reviewer 3 Report

I have found this paper very clear and well written.
The Authors have provided a nicely introduced and well-structured exposition of their material.
The content is described with a sufficient amount of details to understand the topic, results and techniques.
The analysis provided and corresponding figures are fully appropriate to the text and its content.
The list of references to the literature related to this field is also appropriate.

Overall, the content is novel, original and accurate. Results are fully convincing.

Below are a few questions to improve further the manuscript quality:
1) As the average of five experimental results were used as the final result reported for the proposed method in table 5, I find it interesting to add the corresponding standard deviation in this table as well.

2) Now the question for the next tables is how many results are considered for averaging the results obtained with other methods involved in the comparison (when this applies)? Similarly, adding the standard deviation would allow a better appreciation of the relative ranking of methods with fairly similar results.

3) It may also be necessary to specify the limit values of learning and validation rates for which the proposed method falls short in terms of effectiveness. Of course, this question also applies and arises for the other methods compared, but to specify and discuss it at least for the proposed method seems reasonable to me.
